# DIA-Based Proteomic Analysis Reveals MYOZ2 as a Key Protein Affecting Muscle Growth and Development in Hybrid Sheep

**DOI:** 10.3390/ijms25052975

**Published:** 2024-03-04

**Authors:** Dan Zhang, Yaojing Yue, Chao Yuan, Xuejiao An, Tingting Guo, Bowen Chen, Jianbin Liu, Zengkui Lu

**Affiliations:** 1Key Laboratory of Animal Genetics and Breeding on the Tibetan Plateau, Ministry of Agriculture and Rural Affairs, Lanzhou Institute of Husbandry and Pharmaceutical Sciences, Chinese Academy of Agricultural Sciences, Lanzhou 730050, China; zhangdan226@163.com (D.Z.); yueyaojing@caas.cn (Y.Y.); yuanchao@caas.cn (C.Y.); anxuejiao@caas.cn (X.A.); guotingting@caas.cn (T.G.); chenbw202204@163.com (B.C.); 2Sheep Breeding Engineering Technology Research Center of Chinese Academy of Agricultural Sciences, Lanzhou 730050, China

**Keywords:** hybrid sheep, proteomic, MYOZ2, muscle, growth and development

## Abstract

Hybridization of livestock can be used to improve varieties, and different hybrid combinations produce unique breeding effects. In this study, male Southdown and Suffolk sheep were selected to hybridize with female Hu sheep to explore the effects of male parentage on muscle growth and the development of offspring. Using data-independent acquisition technology, we identified 119, 187, and 26 differentially abundant proteins (DAPs) between Hu × Hu (HH) versus Southdown × Hu (NH), HH versus Suffolk × Hu (SH), and NH versus SH crosses. Two DAPs, MYOZ2 and MYOM3, were common to the three hybrid groups and were mainly enriched in muscle growth and development-related pathways. At the myoblast proliferation stage, *MYOZ2* expression decreased cell viability and inhibited proliferation. At the myoblast differentiation stage, MYOZ2 expression promoted myoblast fusion and enhanced the level of cell fusion. These findings provide new insights into the key proteins and metabolic pathways involved in the effect of male parentage on muscle growth and the development of hybrid offspring in sheep.

## 1. Introduction

Hybridization can improve the production performance, survival, and stress resistance of hybrid offspring and is widely used in animal and plant breeding [1]. The hybridization of domestic cattle and bison can not only improve production performance but also facilitate adaptation to harsh living conditions [2]. As the hybrid offspring of a horse and a donkey, mules have both the endurance advantage of donkeys and the physical strength advantage of horses [3]. Omics studies can provide molecular insights into the regulatory networks involved in heterosis or hybrid vigor [4]. Comparative transcriptome analysis of different breeds of sheep identified many genes related to muscle growth and development [5,6,7]. The hybrid offspring of Small Tail Han sheep have higher production performance than pure breeds. By comparing the transcriptome, many genes related to muscle growth and development were found, such as *ITGBL1*, *CPXM2*, and *MRPL1* [8,9,10,11]. Transcriptomics and proteomics analysis of silkworms showed that some genes and proteins that were simultaneously upregulated were involved in various metabolic processes, conferring phenotypic advantages on the hybrid worms [12].

The Hu sheep breed is highly regarded in China for its advantageous features of non-seasonal estrus, multiple births, tolerance to roughage, and suitability for domestic breeding [13,14]. Southdown sheep and Suffolk sheep have the characteristics of early development and easy fattening, and their rams are suitable for hybridization to obtain good growth and carcass characteristics [15,16,17]. Proteomics can be used to systematically characterize large-scale dynamic changes in protein expression [18], contributing to genomics and transcriptomics by deepening our understanding of complex biochemical processes at the molecular level [19]. Wang et al. used proteomics technology to screen the differential abundance proteins (DAPs) of more than 1000 Chinese Merino sheep at different embryonic stages, elucidating mechanisms involved in embryonic skeletal muscle growth, development, and maturation [20]. Screening and identification of seasonal weight loss (SWL) tolerance-related proteins have also been used to improve the production performance of tropical and Mediterranean animals [21]. Additionally, proteomics has been used to study the growth performance of various animals, such as pigs [22], mice [23], and cattle [24,25].

In this study, we used data-independent acquisition (DIA) technology to analyze differences in protein profiles of skeletal muscle samples from Hu × Hu (HH), Southdown × Hu (NH), and Suffolk × Hu (SH) crosses and verified the protein functions at the cellular level. Our main aims were to identify the differences in skeletal muscle proteomes between male parents of different sheep breeds and to explore the mechanism by which male parentage affects muscle growth and development in offspring.

## 2. Results

### 2.1. Qualitative and Quantitative Protein Analyses and Sample Relationship Analysis

The DIA results were analyzed against the DDA reference database to identify proteins. Using the false discovery rate (FDR) ≤ 0.01 filtration standard, 2425 protein groups and 23,833 peptides were identified (Appendix A). For protein quantification, we screened the average peak area of the first three mass spectrometry (MS) peptides with a FDR < 1.0% (Appendix A). The expression levels in any two samples were used to calculate the Pearson correlation coefficient between the two samples. The similarity within and between the three groups of samples was >0.8, with excellent similarity (Figure 1A). Under the conditions of a DAP value of 1, the number of common genes between the three groups was 2623. There were 27 DAPs between HH and SH, NH; 5 DAPs between NH and HH, SH; and 18 DAPs between SH and HH, NH (Figure 1B).

### 2.2. Identification and Analysis of DAPs

The screening thresholds for identification of significant DAPs between the hybrid groups comprised an absolute fold change (FC) value > 1.5 and a corrected *p* value (Qvalue) < 0.05. In the HH versus NH comparison, there were 21 upregulated proteins and 98 downregulated proteins. HH versus SH had 19 upregulated proteins and 168 downregulated proteins, and NH versus SH had 11 upregulated proteins and 15 downregulated proteins (Figure 2A,B and Appendix A). Among the three groups, there were two DAPs: MYOZ2 and MYOM3 (Figure 2C). The relative expression levels of the two DAPs in the groups were HH > SH > NH (Figure 2D).

In the trend analysis, a total of 256 trend genes and 8 gene distribution trends were screened, among which the numbers of proteins assigned versus expected were significant (*p* < 0.05) for two profiles: profile0 and profile1 (Appendix A). Profile0 and profile1 are mainly enriched in fatty acid metabolism and amino acid-related pathways (Appendix A).

### 2.3. Functional Analyses of Identified DAPs

A total of 245 DAPs were enriched in 53 Gene Ontology (GO) terms: 207 DAPs were enriched in 27 biological processes (BP), mainly involving cellular processes, single-organism processes, and metabolic processes; 212 DAPs were enriched in 9 molecular functions (MF), mainly involving binding and catalytic activity, and 204 DAPs were enriched in 19 cellular components (CC), mainly involving cells, cell parts, and organelles (Figure 3A, Appendix A).

The DAPs were enriched in 158 pathways. The Kyoto Encyclopedia of Genes and Genomes (KEGG) pathway enrichment results of the DAPs identified in skeletal muscle samples of the three groups of experimental sheep showed similar differences in the between-group comparisons (Figure 3B, Appendix A). The DAPs of HH and NH were enriched in pathways involved in metabolism, fatty acid degradation, fatty acid metabolism, and fatty acid elongation. The DAPs of HH and SH were enriched in pathways involved in metabolism, fatty acid degradation, fatty acid metabolism, oxidative phosphorylation, and the citric acid cycle. The DAPs of NH and SH were enriched in pathways involved in drug metabolism—other enzymes, complement and coagulation cascades, and pertussis.

### 2.4. Gene Set Enrichment Analysis (GSEA)

In the comparison of HH versus NH, GSEA showed that 34 gene sets were significantly upregulated and 30 gene sets were significantly downregulated. Notably, one of the upregulated gene sets was related to the osteoclast differentiation signaling pathway (Figure 4A), and one of the downregulated gene sets was related to the fatty acid metabolism signaling pathway (Figure 4B). The GSEA of the HH versus SH comparison showed that 19 gene sets were significantly upregulated, one of which was related to the osteoclast differentiation signaling pathway, and 26 gene sets were significantly downregulated, one of which was related to the fatty acid metabolism signaling pathway. In the NH versus SH comparison, GSEA showed that four gene sets were significantly upregulated and 34 gene sets were significantly downregulated, including pathways related to fatty acid degradation and oxidative phosphorylation (Appendix A).

### 2.5. Protein Interaction of MYOZ2 and MYOM3

The basic analysis of sample differences showed that there were two common skeletal muscle DAPs, namely, MYOZ2 and MYOM3, between the three groups of sheep. Protein interaction network analysis showed that there was interaction between MYOZ2 and MYOM3. Between-group comparisons revealed the following: HH versus NH identified 119 DAPs, including 14 protein nodes related to MYOZ2 and MYOM3; HH versus SH identified 187 DAPs, including 20 protein nodes related to MYOZ2 and MYOM3; and NH versus SH identified 26 DAPs, including 6 protein nodes related to MYOZ2 and MYOM3. The DAPs showing interaction with MYOZ2 and MYOM3 were found to be mainly related to muscle growth and development (Figure 5).

### 2.6. The Function of MYOZ2 in Myoblasts

IF of isolated myoblasts showed that the purity of myoblasts reached more than 95%, indicating that they were sufficient for use in the following experiments (Appendix A).

At the myoblast proliferation stage, transfected with the *MYOZ2* overexpression plasmid, the expression of *MYOZ2* was more than 60,000 times that of the negative control group, demonstrating that *MYOZ2* was successfully overexpressed (Figure 6A). After 24 h of *MYOZ2* overexpression, the expression of proliferation marker *PAX7* decreased significantly (*p* < 0.01), and the expression of cell differentiation determining factor *MYOD* increased significantly (*p* < 0.01) (Figure 6B). Transfected with the *MYOZ2* RNA interference plasmid, the expression of *MYOZ2* was 0.04 times that of the negative control group, and the results showed that *MYOZ2* expression was successfully interfered with (Figure 6C). After 24 h of *MYOZ2* with interference, the expression of *PAX7* was significantly increased (*p* < 0.01), and the expression of *MYOD* was significantly decreased (*p* < 0.01) (Figure 6D).

CCK8 results showed that the viability of myoblasts was significantly decreased after overexpression of *MYOZ2*, whereas *MYOZ2* interference had no significant effect on the viability of myoblasts (Figure 6E). The EdU (5-Ethynyl-2-deoxyuridine) assay results showed that the proliferation rate of myoblasts was significantly increased after *MYOZ2* interference (Figure 6F,H) and was significantly decreased after overexpression of *MYOZ2* (Figure 6G,I).

At the stage of myoblast differentiation, after overexpressing *MYOZ2* and inducing differentiation with 2% horse serum (HS) for 3 days, the *MYOZ2* expression level was still more than 20,000 times that of the negative control group (Figure 7A). RT-qPCR results showed that the expression of *MYF5* (cell differentiation early marker gene) was significantly decreased (*p* < 0.01), and the expression of *MYOG* (cell differentiation late marker gene) was not significantly different (*p* > 0.05) compared with the negative control group (Figure 7B). After interfering *MYOZ2* and inducing differentiation with 2% HS for 3 days, the expression efficiency of *MYOZ2* was maintained at about 0.6 in compared with the negative control group (Figure 7C). RT-qPCR results showed that the expression of *MYF5* and *MYOG* was significantly increased (*p* < 0.01) compared with the negative control group (Figure 7D). Compared with the negative control group, the *MYOZ2* overexpression group had a relatively high proportion of myotubes comprising > 5 fused cells, indicating that increasing the expression level of *MYOZ2* increased the degree of myoblast differentiation (Figure 7E,G). Conversely, compared with the negative control group treatment group, the *MYOZ2* interference group had a relatively low proportion of myotubes comprising > 5 fused cells, indicating that reducing the expression level of *MYOZ2* reduced the degree of myoblast differentiation (Figure 7F,H).

## 3. Discussion

Generally, the expression level of a gene affects the abundance and function of its protein, but multiple post-transcriptional and post-translational mechanisms also affect protein abundance and function [19]. The purpose of differential analysis is to screen for proteins with significant changes in abundance between comparison groups. In this study, comparative proteomics analysis of different paternal hybrid sheep breeds was performed to identify biomarkers related to muscle development in the hybrids. A total of 2778 proteins were identified in muscle tissue, including key proteins related to fat synthesis and muscle development. Meat quality in sheep is associated with muscle growth and development and intramuscular fat deposition. The DAPs identified in this study may help to reveal the genetic mechanism of sheep muscle growth and development using label-free proteomics strategies. Further comparative proteomic analysis of HH, NH, and SH may also reveal potential genetic differences in sheep breed formation.

Two DAPs, namely, MYOZ2 and MYOM3, were identified in the comparison of three groups of experimental sheep. MYOZ2 belongs to the myozenin family, which binds calcineurin [26]. Calcineurin is a calcium/calmodulin-dependent serine, threonine phosphatase that plays an important role in transducing calcium-dependent signals in a variety of cell types [27]. MYOZ2 expression effectively inhibits calcineurin activity and thus helps in the regulation of muscle fiber differentiation [28]. In experiments on Qinchuan cattle, *MYOZ2* knockdown was found to inhibit the differentiation of bovine myoblasts [29]. MYOM3 is the myofibrillar structural protein myomesin-3 [30]. MYOM3 may link the intermediate filament cytoskeleton to the M-disk of the myofibrils in striated muscle and is thought to be involved in muscle function and repair [30,31,32], suggesting that the expression levels of MYOZ2 and MYOM3 are associated with muscle growth and development. In the HH versus NH and HH versus SH comparisons, both MYOZ2 and MYOM3 were upregulated, while in the NH versus SH comparison, both MYOZ2 and MYOM3 were downregulated. This result suggests that there are differences in muscle growth and development among the offspring of these three different paternal lines.

Muscle proliferation and differentiation is a biological process that is strictly regulated by a large number of proteins [33,34]. Skeletal muscle growth and development are essential for quality meat production from livestock [35]. MYOZ2 is a muscle tissue-specific, intracellular binding protein that has involvement in connecting Z-line proteins and transmitting calcineurin signals to sarcomeres and plays an important role in skeletal muscle and myocardium [36,37]. Studies have shown that *MYOZ2* is regulated by *MEF2A* (myocyte enhancer factor 2A), which in turn affects myoblast differentiation [38]. Prior to this study, there were no reports on the proliferation and differentiation of sheep muscle by MYOZ2. To determine the regulatory mechanism of MYOZ2 on the proliferation and differentiation of sheep muscle, *MYOZ2* was overexpressed and silenced in Hu sheep myoblasts. We found that *MYOZ2* negatively regulated *PAX7* in myoblasts in the proliferative stage and positively regulated *PAX7* in myoblasts in the differentiation stage, suggesting that the *PAX7* gene may be a key regulator of the entry of myogenic progenitor cells into skeletal muscle, preventing premature differentiation [39]. CCK8 results in this study showed that *MYOZ2* overexpression reduced myoblast viability and promoted apoptosis. EdU assay results showed that *MYOZ2* overexpression inhibited myoblast proliferation, whereas *MYOZ2* silencing promoted myoblast proliferation. This suggested that *MYOZ2* may regulate genes associated with proliferation, which in turn regulates myoblast proliferation. This study also showed that *MYOZ2* negatively regulated *MYF5* and *MYOD1* in myoblasts in the cell differentiation phase. However, it had only a small regulatory effect on *MYOG*. *MYF5* and *MYOD1* are myogenic determinant genes expressed in the early stage of differentiation. Because *MYF5* is expressed before *MYOD* [40], *MYOZ2* has a greater regulatory effect on *MYF5* than it does on *MYOD*. *MYOG* is a marker gene of the late stage of cell differentiation [41], and *MYOZ2* has less regulatory effect on *MYOG* when differentiation is 3 d. Genes corresponding to proteins interacting with MYOZ2 co-enriched two BPs of muscle system process and anatomical structure development [42,43,44,45,46,47], among which TNNT1 and NRAP were enriched in cell differentiation [46], and the MF of MYL3 and MYL6 are structural constituents of muscle [48,49]. The results of inducing the differentiation of achievement cells showed that the experimental group with a high *MYOZ2* expression level was more differentiated. In contrast, the *MYOZ2* silent expression group was less differentiated. The results of this study showed that *MYOZ2* expression promoted myoblast differentiation and possibly regulated differentiation-related genes, thereby regulating myoblast differentiation.

## 4. Materials and Methods

### 4.1. Animals and Sample Collection

Ninety rams were divided into three groups: NH, SH, and HH (*n* = 30 rams per group). Six rams in each group were randomly selected for slaughter at the age of 12 months. The within-group body weight of each sheep was similar (HH: 52.83 ± 4.06 kg, NH: 58.06 ± 3.58 kg, and SH: 65.51 ± 4.64 kg). These 18 test sheep were slaughtered 24 h after food deprivation, but had free access to water. The left longissimus dorsi was obtained from the 18 test sheep within 30 min post slaughter, and external fat and connective tissue were removed before subsequent analyses. The instruments used in this study are described in Appendix A.

### 4.2. Protein Extraction and Digestion

Samples were transferred into lysis buffer (2% SDS, 7 M urea, 1 mg/mL protease inhibitor cocktail), and homogenized for 3 min (bacteria 5 min; tissue 180 s three times) in ice using an ultrasonic homogenizer. The homogenate was centrifuged at 15,000 rpm for 15 min at 4 °C, and the supernatant was collected.

The BCA Protein Assay Kit was used to determine the protein concentration of the supernatant. For this, 50 μg proteins extracted from cells were suspended in 50 μL solution, reduced by adding 1 μL 1 M dithiothreitol at 55 °C for 1 h, and alkylated by adding 5 μL of 20 mM iodoacetamide in the dark at 37 °C for 1 h. Then, the sample was precipitated using 300 μL of prechilled acetone at −20 °C overnight. The precipitate was washed twice with cold acetone and then resuspended in 50 mM ammonium bicarbonate. Finally, the proteins were digested with sequence-grade modified trypsin (Promega, Madison, WI, USA) at a substrate/enzyme ratio of 50:1 (*w*/*w*) at 37 °C for 16 h.

### 4.3. High-pH Reverse-Phase Separation

The peptide mixture was re-dissolved in buffer A (buffer A: 20 mM ammonium formate in water, pH 10.0, adjusted with ammonium hydroxide), and then fractionated by high pH separation using the Ultimate 3000 system (ThermoFisher Scientific, Waltham, MA, USA) connected to a reverse-phase column (XBridge C18 column, 4.6 mm × 250 mm, 5 μm, Waters Co., Milford, MA, USA). High pH separation was performed using a linear gradient, starting from 5% B to 45% B in 40 min (B: 20 mM ammonium formate in 80% ACN, pH 10.0, adjusted with ammonium hydroxide). The column was re-equilibrated at the initial condition for 15 min. The column flow rate was maintained at 1 mL/min, and the column temperature was maintained at 30 °C. Ten fractions were collected, and each fraction was dried in a vacuum concentrator for the next step.

### 4.4. DDA: Nano-HPLC-MS/MS Analysis

The peptides were re-dissolved in 30 μL of solvent A (A: 0.1% formic acid in water) and analyzed by on-line nanospray LC-MS/MS on an Orbitrap Fusion Lumos coupled to the EASY-nLC 1200 system (Thermo Fisher Scientific, Waltham, MA, USA). The 3 μL peptide sample was loaded onto the analytical column (Acclaim PepMap C18, 75 μm × 25 cm) and separated with a 120 min gradient from 5% to 35% B (B: 0.1% formic acid in ACN). The column flow rate was maintained at 200 nL/min with a column temperature of 40 °C. The electrospray voltage of 2 kV versus the inlet of the mass spectrometer was used. The mass spectrometer was run in data-dependent acquisition mode, and it was automatically switched between MS and MS/MS modes. The parameters were: (1) MS: scan range (*m*/*z*) = 350–1200; resolution = 120,000; AGC target = 400,000; maximum injection time = 50 ms; and filter dynamic exclusion: exclusion duration = 30 s. (2) HCD-MS/MS: resolution = 15,000; AGC target = 50,000; maximum injection time = 35 ms; and collision energy = 32.

### 4.5. Database Search

Raw data from DDA were processed and analyzed by Spectronaut X (Biognosys AG, Zurich, Switzerland) with default settings to generate an initial target list. Spectronaut was set up to search the sheep database along with the contaminant database, assuming trypsin as the digestion enzyme. Carbamidomethyl (C) was specified as the fixed modification. Oxidation (M) was specified as the variable modification. The Qvalue (FDR) cutoff on precursor and protein levels was applied at 1%.

### 4.6. DIA: Nano-HPLC-MS/MS Analysis

The peptides were re-dissolved in 30 μL of solvent A (A: 0.1% formic acid in water) and analyzed by on-line nanospray LC-MS/MS on an Orbitrap Fusion Lumos coupled to the EASY-nLC 1200 system (Thermo Fisher Scientific, Waltham, MA, USA). The 3 μL peptide sample was loaded onto the analytical column (Acclaim PepMap C18, 75 μm × 25 cm) with a 120 min gradient from 5% to 35% B (B: 0.1% formic acid in ACN). The column flow rate was maintained at 200 nL/min with a column temperature of 40 °C. The electrospray voltage of 2 kV versus the inlet of the mass spectrometer was used.

The mass spectrometer was run in data-independent acquisition mode, and it was automatically switched between MS and MS/MS modes. The parameters were: (1) MS: scan range (*m*/*z*) = 350–1200; resolution = 120,000; AGC target = 1 × 10^6^; and maximum injection time = 50 ms. (2) HCD-MS/MS: resolution = 30,000; AGC target = 1 × 10^6^; collision energy = 32; and stepped CE = 5%. (3) DIA was performed with a variable isolation window, and each window overlapped 1 *m*/*z*, and the window number was 60.

### 4.7. DIA Data Acquisition and Analysis

The data-dependent acquisition (DDA) library had 2806 protein groups and 28,531 peptides (Appendix A). Raw data from DIA were processed and analyzed by Spectronaut X (Biognosys AG, Switzerland) with default parameters. The retention time prediction type was set to dynamic iRT. Data extraction was determined by Spectronaut X based on the extensive mass calibration. Spectronaut Pulsar X determines the ideal extraction window dynamically, depending on iRT calibration and gradient stability. The Qvalue (FDR) cutoff on precursor and protein levels was applied at 1%. Decoy generation was set to mutated, which is similar to scrambled but will only apply a random number of AA position swamps (min = 2, max = length/2). All selected precursors passing the filters were used for quantification. The average top 3 filtered peptides that passed the 1% Qvalue cutoff were used to calculate the major group quantities. After the Student’s *t*-test, different expressed proteins were filtered if their Qvalue was <0.05 and their absolute AVG log2 ratio was >0.58 (Appendix A).

### 4.8. Protein Quantitative Normalization Treatment

Qualitative protein analysis requires the identification of precursor thresholds of 1.0% FDR and protein thresholds of 1.0% FDR at the peptide and protein levels, respectively. The local normalization method in Pulsar was used to normalize the peak intensity of the overall sample. The signal intensities of most samples were substantially the same (Appendix A). The average of the peak areas of the first 3 MS1 peptides with an FDR of less than 1.0% was screened for protein quantification (Appendix A, Appendix A).

### 4.9. Bioinformatics Analyses

Proteins were annotated against the GO, KEGG, and COG/KOG databases to obtain their functions. Significant GO functions and pathways were examined within differentially expressed proteins with a Qvalue ≤ 0.05. The interaction relationship in the STRING protein interaction database (http://string-db.org accessed on 15 July 2021) was used to analyze the differential protein interaction network. Gene Set Enrichment Analysis (GSEA, http://software.broadinstitute.org/gsea/index.jsp accessed on 15 July 2021) was used to perform GSEA analysis on the KEGG pathway and screen pathways related to muscle growth and development from the database.

### 4.10. Isolation, Culture, and Identification of Myoblasts

Hu sheep were euthanized at 1 day of age, and leg muscles were removed for the isolation of myoblasts. As previously described, myoblasts were isolated by two-step digestion with type IV collagenase (Gibco, Grand Island, NY, USA) and 0.25% trypsin-EDTA (Gibco, Grand Island, NY, USA) [50] and purified by the removal of fibroblasts using differential adhesion. After achieving a stable cell state, immunofluorescence (IF) with antibodies against PEX7, MYOD, and Desmin was used to identify myoblasts (Appendix A).

### 4.11. Construction and Transfection

The *MYOZ2* overexpression plasmid was constructed by inserting the *MYOZ2* coding sequence into the pEX3 vector (Appendix A). The interference plasmid was constructed by inserting the target sequence (CTGGCAGACGGTCCTTTAATA) into the pGPU6/green fluorescent protein (GFP)/neomycin (Neo) vector (Appendix A). The expression level of MYOZ2 was interfered with by using DNA plasmids that transcribed siRNA. The 4th passage of myoblast cells is cultured for 24 h after that the overexpressing and interfering plasmids were transfected with the Lipofectamine 3000 Transfection Kit (Thermo Fisher Scientific, Waltham, MA, USA) at a ratio of 5 μL Lipofectamine 3000:2500 ng plasmid. Unlike the transfection of pEX3 into cells experiment, when transfecting siRNA into cells, the P3000™ reagent (Thermo Fisher Scientific, Waltham, MA, USA) is not added while diluting the siRNA.

### 4.12. Quantitative Reverse Transcriptase PCR (RT-qPCR)

After a 24 h transfection, cell samples were collected using cell lysis buffer (Gbico, Grand Island, NY, USA). RNA was extracted by the TRIzol method and reverse transcribed using a one-step reverse transcription kit (Transgen, Haidian District, Beijing, China). The expression levels of the myoblast proliferation marker genes *PAX7* and *MYOD* and the myoblast differentiation marker genes *MYOG* and *MYF5* were detected by RT-qPCR (Transgen) (Appendix A).

### 4.13. Cell Proliferation Assay

5-Ethynyl-2-deoxyuridine (EdU, Ribobio, Guangzhou, China) was used to detect newly synthesized DNA of the cells, and the cell proliferation rate was detected by double labeling with a nuclear marker (Hoechest 33342).

### 4.14. Cell Counting Kit-8 (CCK8)

Cells were seeded in 96 well plates and incubated in an incubator for 24 h, followed by transfection experiments. At 0 h, 24 h, 48 h, and 72 h after transfection, 10 μL of CCK8 solution was added to each well, and plates were incubated at 37 °C in 5% CO_2_ for 2 h. A blank set was prepared with six replicates per treatment set. The optical density (OD) value was determined using a microplate reader. Results are expressed as the average of the treatment group minus the blank group (CCK8 Kit from Thermo Fisher Scientific).

### 4.15. IF

Cells were seeded in 6 well plates with 3 replicates per treatment, and incubated in an incubator for 24 h prior to transfection experiments. Post-transfection, cells were incubated for an additional 24 h, then fixed with 4% paraformaldehyde for 20 min, permeabilized with 3% TRIton X-100 (Sigma, St. Louis, MO, USA) for 10 min, blocked with buffer containing 5% fetal bovine serum (Gbico) for 1 h, incubated overnight at 4 °C with primary antibody (myosin heavy chain 3, MYH3), and then again incubated with secondary antibody (Cyanine 3, Cy3) at room temperature for 1 h. Observations were made using a ZEISS LSM800 confocal laser scanning microscope (Plan APOCHROMAT 10×/0.45, Carl Zeiss, Oberkochen, German), and images were processed using ZEN software (ZEISS ZEN 2.3, Carl Zeiss, Oberkochen, German).

### 4.16. Statistical Analysis

To analyze the data, MS Excel was used for simple statistics, and SPSS 23.0 was used for one-way analysis of variance. Results are presented as the mean ± standard error. A *p* value > 0.05 was considered insignificant, 0.01 ≤ *p* ≤ 0.05 was considered significant, and *p* < 0.01 was considered extremely significant.

## 5. Conclusions

In summary, a large number of DAPs were identified by proteomic analysis of hybrid sheep muscles. Hybrids with different male parentages exhibited differential expression of muscle development-related proteins and pathways, including oxidative phosphorylation. At the cellular level, MYOZ2 was shown to be a positive regulator of hybrid myoblast differentiation. Overall, these findings deepen our understanding of the key proteins and pathways involved in muscle growth in the hybrid sheep population.

## Figures and Tables

**Figure 1 ijms-25-02975-f001:**
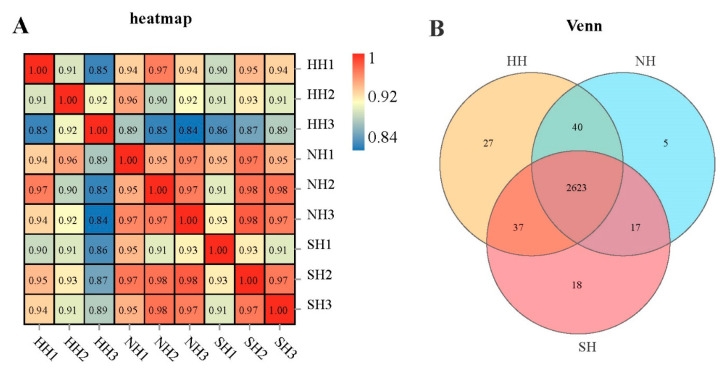
Sample relationship analysis. (**A**) Correlation heat map is of the three groups of samples correlation heat map. (**B**) The Venn diagram of proteins among three groups of samples.

**Figure 2 ijms-25-02975-f002:**
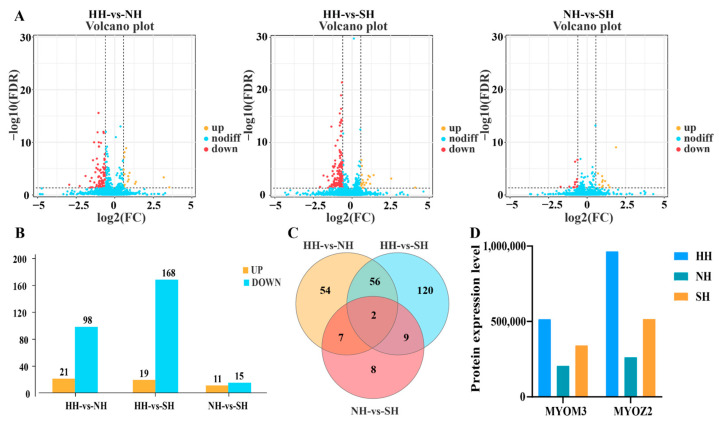
Identification and analysis of DAPs in the three groups of hybrid sheep. (**A**) The difference comparison volcano plot. (**B**) The overall statistical chart of differences. (**C**) The Venn diagram. (**D**) The expression level histogram of MYOZ2 and MYOM3 muscles. This figure shows the expression levels of MYOZ2 and MYOM3 in muscle tissue samples used for DIA analysis.

**Figure 3 ijms-25-02975-f003:**
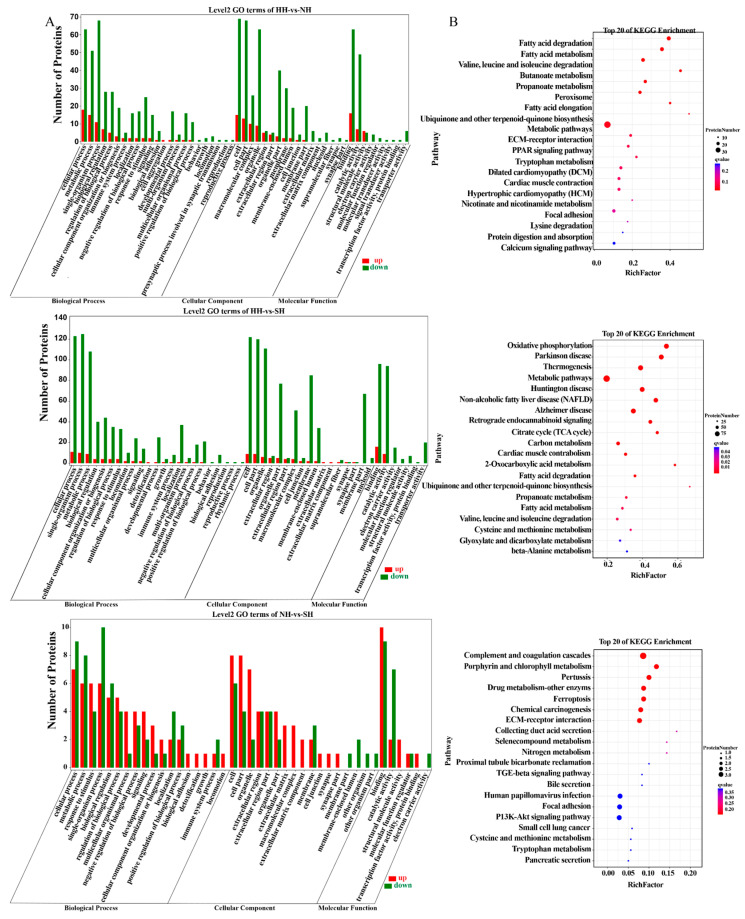
Functional enrichment analysis of DAPs. (**A**) Bar chart of the GO enrichment classification of DAPs. (**B**) KEGG enrichment bubble diagram of DAPs.

**Figure 4 ijms-25-02975-f004:**
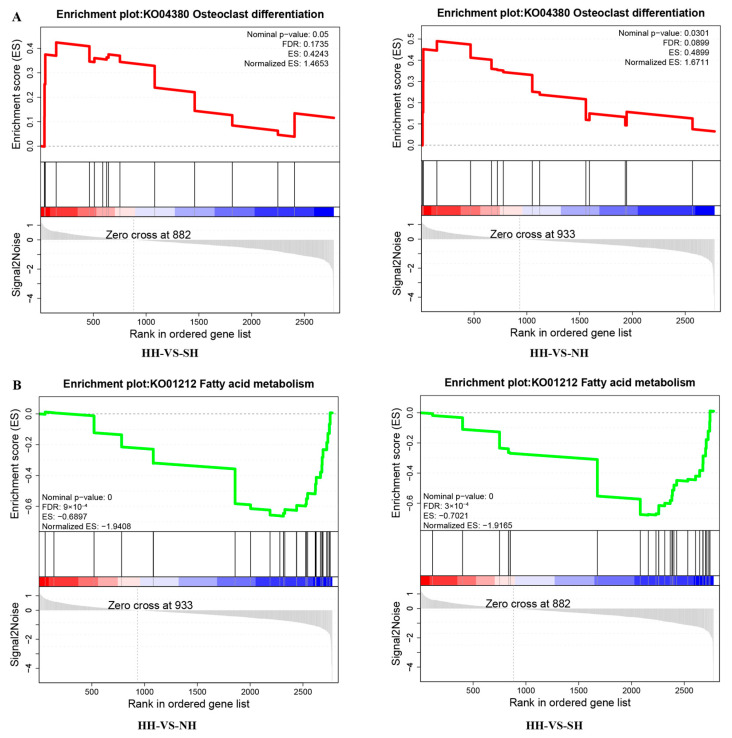
GSEA analysis of DAPs. (**A**) Enrichment lot of osteoclast differentiation signaling pathway, the positive ES (enrichment score, red curve) indicates that the gene set is enriched at the top of the list. (**B**) Enrichment lot of fatty acid metabolism signaling pathway, the negative ES (green curve) indicates that the gene set is enriched at the bottom of the list.

**Figure 5 ijms-25-02975-f005:**
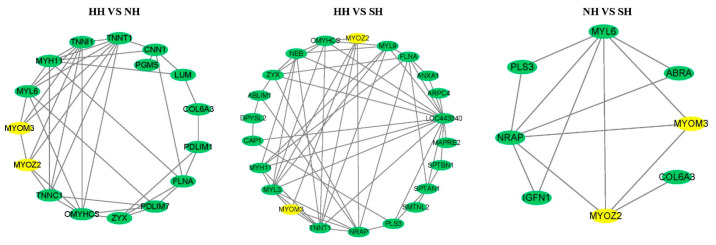
Protein interaction network of muscle growth and development. The yellow marks highlight the presence of MYOZ2 and MYOM3 in all three groups of protein interaction networks. The green marker highlights the DAPs associated with muscle growth and development in all three groups of protein interaction networks.

**Figure 6 ijms-25-02975-f006:**
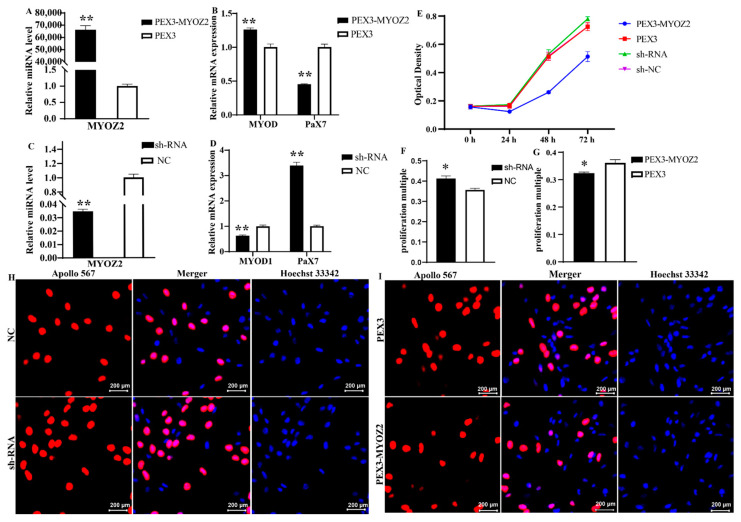
Effect of *MYOZ2* on the proliferation of myoblasts. (**A**) Determination of *MYOZ2* overexpression efficiency. (**B**) Determination of marker gene expression efficiency in myoblasts under *MYOZ2* overexpression. (**C**) Determination of *MYOZ2* interference efficiency. (**D**) Determination of marker gene expression efficiency in myoblasts under *MYOZ2* interference. (**E**) CCK8 results of myoblasts. (**F**–**I**) EdU assay of *MYOZ2* interference (**F**,**H**) and *MYOZ2* overexpression (**G**,**I**) in myoblasts. * indicates significant differences between comparison groups, ** indicates extremely significant differences between comparison groups.

**Figure 7 ijms-25-02975-f007:**
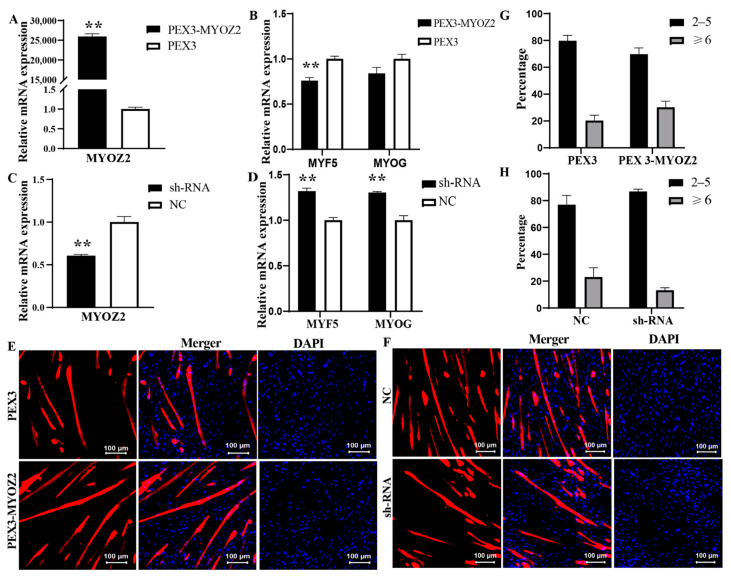
Effect of *MYOZ2* on the differentiation of myoblasts. (**A**) Determination of *MYOZ2* overexpression efficiency in myoblasts. (**B**) Determination of marker gene expression efficiency under *MYOZ2* overexpression in myoblast. (**C**) Determination of *MYOZ2* interference efficiency in myoblast. (**D**) Determination of marker gene expression efficiency under *MYOZ2* interference in myoblast. (**E**,**F**) IF of *MYH3* in myotubes under overexpression of *MYOZ2* (**E**) or interference with *MYOZ2* (**F**). (**G**,**H**) Statistical analysis of the number of fused cells in myotubes of differentiating myoblast cultures under overexpression of *MYOZ2* (**G**) or interference with *MYOZ2* (**H**). ** indicates extremely significant differences between comparison groups.

## Data Availability

Data will be made available on request.

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
