# Peer review of "DIA-Based Proteomic Analysis Reveals MYOZ2 as a Key Protein Affecting Muscle Growth and Development in Hybrid Sheep"

_ijms, 2024, doi:10.3390/ijms25052975_

Round 1

Reviewer 1 Report

Comments and Suggestions for Authors

In this manuscript, the Authors used data-independent acquisition (DIA) proteomics to analyze the protein expression changes in skeletal muscle samples from Hu × Hu (HH), Southdown × Hu (NH), and Suffolk × Hu (SH) sheep crosses. In particular, the Authors were able to investigate the differences in skeletal muscle proteomes between male parents of different sheep breeds and explore the mechanism by which male parentage affects muscle growth and development of offspring. The Authors found that MYOZ2 expression at the myoblast proliferation stage decreased cell viability and inhibited proliferation. At the myoblast differentiation stage, MYOZ2 expression was found to promote myoblast fusion. The Authors have done well to investigate and discuss the interesting functions of MYOZ2 in myoblasts.

Overall, this manuscript provides needed omics insights and new proteomics approaches for investigating the skeletal muscle changes of hybrid sheep. DIA methods are an emerging and powerful proteomics tool and wider adoption of such proteomics methods are sure to enrich the broader skeletal muscle community. The conclusions drawn are supported by the data, the citations are appropriate, and the results are of general interest to the skeletal muscle and broader omics community. However, there are some minor data presentation and clarity improvements that should be made and will strengthen the final manuscript. Therefore, I recommend the publication of this manuscript once the specific comments included below are addressed:

Specific comments to be addressed:

1. For Figure 1B: Do you mean “Venn” diagram?

2. Figure 2: Please clarify what is meant by “protein expression level”. Is this the normalized intensity obtained post DIA processing? If so, is this at the MS1 or MS2 level? These key details should be stated. Also, as a general point, it is challenging to refer to the absolute intensity values obtained by a DIA method, especially without any quantitative follow-up. I would caution to make sure the wording is clear here.

3. For Figure 3: Please increase the font size on the labels. It is currently unreadable.

4. Figure 4 and in general regarding the Gene Set Enrichment Analysis (GSEA): It is not very clearly stated how these results were obtained. More information should be provided in the “Bioinformatics Analysis” section of the Methods.

5. For Figure 5: It is not clear what the color coding of the genes in the interaction network graph means.

Author Response

Reviewer #1:

Comments and Suggestions for Authors

In this manuscript, the Authors used data-independent acquisition (DIA) proteomics to analyze the protein expression changes in skeletal muscle samples from Hu × Hu (HH), Southdown × Hu (NH), and Suffolk × Hu (SH) sheep crosses. In particular, the Authors were able to investigate the differences in skeletal muscle proteomes between male parents of different sheep breeds and explore the mechanism by which male parentage affects muscle growth and development of offspring. The Authors found that MYOZ2 expression at the myoblast proliferation stage decreased cell viability and inhibited proliferation. At the myoblast differentiation stage, MYOZ2 expression was found to promote myoblast fusion. The Authors have done well to investigate and discuss the interesting functions of MYOZ2 in myoblasts.

Overall, this manuscript provides needed omics insights and new proteomics approaches for investigating the skeletal muscle changes of hybrid sheep. DIA methods are an emerging and powerful proteomics tool and wider adoption of such proteomics methods are sure to enrich the broader skeletal muscle community. The conclusions drawn are supported by the data, the citations are appropriate, and the results are of general interest to the skeletal muscle and broader omics community. However, there are some minor data presentation and clarity improvements that should be made and will strengthen the final manuscript. Therefore, I recommend the publication of this manuscript once the specific comments included below are addressed:

We appreciate your suggestions. According to your suggestions, we revised the contents of the manuscript and marked the modified parts in red.

Q1. For Figure 1B: Do you mean “Venn” diagram?

 Response: Thank you for pointing out the error. We have fixed this error in the new version of the article in line 76.

Q2. Figure 2: Please clarify what is meant by “protein expression level”. Is this the normalized intensity obtained post DIA processing? If so, is this at the MS1 or MS2 level? These key details should be stated. Also, as a general point, it is challenging to refer to the absolute intensity values obtained by a DIA method, especially without any quantitative follow-up. I would caution to make sure the wording is clear here.

Response: Thank you for your precious comments. The result in Figure 2 is the normalized intensity obtained post DIA processing and this at the MS1 level. The average of the peak areas of the first 3 MS1 peptides with an FDR of less than 1.0% was screened for protein quantification. we have added the following sentences to this paragraph of Materials and Methods.

“4.8 Protein quantitative normalization treatment

Qualitative protein analysis requires the identification of Precursor Threshold 1.0% FDR and Protein Threshold 1.0% FDR at the peptide and protein levels, respectively. The local normalization method in Pulsar was used to normalize the peak intensity of the overall sample. The signal intensity of most samples was substantially the same (Figure S1B). The average of the peak areas of the first 3 MS1 peptides with an FDR of less than 1.0% was screened for protein quantification (Table S6, Figure S1C).” (lines 346-352, page 10)

Q3. For Figure 3: Please increase the font size on the labels. It is currently unreadable.

 Response: Thank you very much for pointing out this problem. We've made the text in Figure 3 larger, bolder.

Q4. Figure 4 and in general regarding the Gene Set Enrichment Analysis (GSEA): It is not very clearly stated how these results were obtained. More information should be provided in the “Bioinformatics Analysis” section of the Methods.

Response: Thank you for your comment. According to your suggestion, the redundant part about GSEA analysis method has been deleted, and we have added relevant content in Materials and Methods as follows:

“Use Gene Set Enrichment analysis (GSEA, http://software.broadinstitute.org/gsea/index.jsp) performed GSEA analysis on the KEGG Pathway and screened pathways related to muscle growth and development from the database.” (lines 358-360, page 10)

Q5. For Figure 5: It is not clear what the color coding of the genes in the interaction network graph means.

Response: Thank you for pointing out the problem. In order for readers to better understand the contents of the figure 5, we added figure notes as follows:

“The yellow color is to emphasize the presence of MYOZ2 and MYOM3 in all three groups of protein interaction networks.” (lines 144-146, page 6)

Reviewer 2 Report

Comments and Suggestions for Authors

In the paper named “DIA-based proteomic analysis reveals MYOZ2 as a key protein affecting muscle growth and development in hybrid sheep” author make a good proteomic study to search different protein expression of skeletal muscle samples from Hu × Hu (HH), Southdown × Hu (NH), and Suffolk × Hu (SH) crosses. Therefore author perform a validation the protein function of the interest proteins making assays as cellular levels.

Only minor questions are required

1)     Author do not mention how they perform a protein library to make the proteomic assay

2)     How author make the figure 2D the protein expression levels are very high. Have obtained this protein expression levels.

3)     Author do not provide information about the proteins upregulted and downregulated, a table including the protein name, uniprot code, p value and fold change it will be necessary.

4)     In the same sense author perform the functional analyse using both up and down regulated proteins however the information about the pathways up and down regulated only is possible using by separated the up and the down regulated proteins.

5)     Figure 3 is difficult to read

6)     Authors see the trasnfeccion levels using the relative miRNA levels, have author data about the protein levels? A western showing the overexpresion will be necessary in order to confirm that in the cells there are protein

7)     The trasnfection with the MYOZ2 RNA interference was performed in the same cells that the overexpresion of MYOZ2. Have author a model with the high levels of MYOZ2? Have author data about the protein levels?

8)     Figure 6E is missing in the text

9)     In figure 7 similar to figure 6 a protein expression levels will be necessary

10) It seems that author perform firs a overexpresion of MYOZ2 (as a transitory trasnfection) and over this cells they perform a miRNA interference, this late experiment is missing in methods. Please can author explain how they perform the interference? It is very confusing the lipofectamin do not make interference with the siRNA? Have author see apoptosis in these cells?

Author Response

Reviewer #2:

Comments and Suggestions for Authors:

In the paper named “DIA-based proteomic analysis reveals MYOZ2 as a key protein affecting muscle growth and development in hybrid sheep” author make a good proteomic study to search different protein expression of skeletal muscle samples from Hu × Hu (HH), Southdown × Hu (NH), and Suffolk × Hu (SH) crosses. Therefore author perform a validation the protein function of the interest proteins making assays as cellular levels.

Thank you very much for your professional review of our articles. According to your suggestion, we have carefully revised the previous manuscript. Revisions in the text are shown in red.

Q1. Author do not mention how they perform a protein library to make the proteomic assay

Response: Thank you for pointing out the problem. we have added the following sentences to this paragraph of Materials and Methods.

4.4 DDA: nano-HPLC-MS/MS Analysis

The peptides were re-dissolved in 30 μL solvent A (A: 0.1% formic acid in water) and analyzed by on-line nanospray LC-MS/MS on an Orbitrap Fusion Lumos coupled to EASY-nLC 1200 system (Thermo Fisher Scientific, MA, USA). 3 μL peptide sample was loaded onto the analytical column (Acclaim PepMap C18, 75 μm x 25 cm) and separated with a 120-min gradient, from 5% to 35% B (B: 0.1% formic acid in ACN). The column flow rate was maintained at 200 nL/min with the column temperature of 40°C. The electrospray voltage of 2 kV versus the inlet of the mass spectrometer was used. The mass spectrometer was run under data dependent acquisition mode, and automatically switched between MS and MS/MS mode. The parameters was: (1) MS: scan range (m/z)=350–1200; resolution=120,000; AGC target=400000; maximum injection time=50 ms; Filter Dynamic Exclusion: exclusion duration=30s; (2) HCD-MS/MS: resolution=15,000; AGC target=50000; maximum injection time=35 ms; collision energy=32.

4.5 Database search

Raw Data of DDA were processed and analyzed by Spectronaut X (Biognosys AG, Switzerland) with default settings to generate an initial target list. Spectronaut was set up to search the database of sheep along with contaminant database assuming trypsin as the digestion enzyme. Carbamidomethyl (C) was specified as the fixed modification. Oxidation (M) was specified as the variable modifications. Qvalue (FDR) cutoff on precursor and protein level was applied 1%.” (lines 299-318, page 9-10)

Q2. How author make the figure 2D the protein expression levels are very high. Have obtained this protein expression levels.

Response: Thank you for pointing out the problem. This figure shows the expression levels of MYOZ2 and MYOM3 in muscle tissue samples used for DIA analysis. In order for readers to better understand the contents of the figure 5, we have added figure notes as follows:

“This figure shows the expression levels of MYOZ2 and MYOM3 in muscle tissue samples used for DIA analysis” (lines 94-95, page 3)

Q3. Author do not provide information about the proteins upregulted and downregulated, a table including the protein name, uniprot code, p value and fold change it will be necessary.

Response: Thank you for your precious comment and advice. we have added the supplementary table form to the supplementary materials and cited in the revised manuscript. (lines 83, page 2)

Q4. In the same sense author perform the functional analyse using both up and down regulated proteins however the information about the pathways up and down regulated only is possible using by separated the up and the down regulated proteins.

Response: Thank you for pointing out this problem. We have added the information about up-regulation and down-regulation of protein in the attachment.

Q5. Figure 3 is difficult to read

Response: Thank you very much for pointing out this problem. We've made the text in Figure 3 larger, bolder.

Q6. Authors see the trasnfeccion levels using the relative miRNA levels, have author data about the protein levels? A western showing the overexpresion will be necessary in order to confirm that in the cells there are protein

Response: Thank you very much for pointing out this important issue. We agree with your comments that preliminary experiments are also necessary. Unfortunately, we didn't find the right antibody.

Thank you very much for pointing out this important issue. We agree with your comments that preliminary experiments are also necessary. Unfortunately, we didn't find the right antibody.

Previous studies have shown that PaX7, MYF5, MYOG and MYOD are key factors involved in the regulation of muscle growth and development. The transformation of H3K4me3 and H3K27me3 on the Pax7 promoter is a key step in myoblast differentiation [1]. The results of this study showed that the expression of key factors changed by regulating the expression of MYOZ2. It can be proved from the transcriptome level that changes in MYOZ2 expression can affect myoblast differentiation.

[1]           Palacios, D.; Mozzetta, C.; Consalvi, S.; Caretti, G.; Saccone, V.; Proserpio, V.; Marquez, V.E.; Valente, S.; Mai, A.; Forcales, S.V.; Sartorelli, V.; Puri, P.L. TNF/p38α/polycomb signaling to Pax7 locus in satellite cells links inflammation to the epigenetic control of muscle regeneration. Cell Stem Cell, 2010, 7: 455-469. 10.1016/j.stem.2010.08.013

Q7. The trasnfection with the MYOZ2 RNA interference was performed in the same cells that the overexpresion of MYOZ2. Have author a model with the high levels of MYOZ2? Have author data about the protein levels?

Response: We are very sorry for causing your misunderstanding. MYOZ2 interference and MYOZ2 overexpression are treated separately, and we both use 4 generations of untreated primary myoblasts. At the same time, we do not have high-level model and protein data for MYOZ2. But we did all the transfection experiments 24 hours after myoblast attachment. For better understanding of the readers, we have revised the description in the revised manuscript (lines 374-377, page 11)

Q8. Figure 6E is missing in the text.

Response: Thank you for pointing out that we misquote images. We have corrected this error in the revised manuscript to read:

“CCK8 results showed that the viability of myoblasts was significantly decreased after overexpression of MYOZ2, whereas MYOZ2 interference had no significant effect on the viability of myoblasts (Figure 6E). The EdU (5-Ethynyl-2-deoxyuridine) assay results showed that the proliferation rate of myoblasts was significantly increased after MYOZ2 interference (Figure 6F, Figure 6H), and was significantly decreased after overexpression of MYOZ2 (Figure 6G, Figure 6I)” (lines 162-165, page 6)

Q9. In figure 7 similar to figure 6 a protein expression levels will be necessary

Response: It is also because there is no suitable antibody for the verification of protein level.

Q10. It seems that author perform firs a overexpresion of MYOZ2 (as a transitory trasnfection) and over this cells they perform a miRNA interference, this late experiment is missing in methods. Please can author explain how they perform the interference? It is very confusing the lipofectamin do not make interference with the siRNA? Have author see apoptosis in these cells?

Response: Thank you for your comment. Interference and overexpression of MYOZ2 were treated separately, and 4 generations of primary myoblasts without any treatment were used for transfection, rather than overexpression and then interference.

As for your doubt that siRNA will not be affected by your lipid, we used DNA plasmid capable of transcribing siRNA for transfection experiment. We are sorry that this content was not pointed out in the manuscript, and we have added this part of content to the Materials and Methods of the revised manuscript, as follows:

4.11 Construction and transfection

The MYOZ2 overexpression plasmid was constructed by inserting the MYOZ2 coding sequence into the pEX3 vector (Supplement table 4). The interference plasmid was constructed by inserting the target sequence (CTGGCAGACGGTCCTTTAATA) into the pGPU6/green fluorescent protein (GFP)/neomycin (Neo) vector(Supplement table 5). And interfering with MYOZ2 expression levels using DNA plasmids that transcribe siRNA. Myoblasts were cultured to passage 4 and transfected with a Lipofectamine 3000 transfection kit (Thermo Fisher Scientific) at the ratio of 5 μL Lipofectamine 3000 to 2500 ng plasmid. Unlike the transfection of pEX3 into cells experiment, when transfecting siRNA into cells, the P3000™ reagent is not added while diluting the siRNA.” (lines 368-378, page 11)

We did not use flow cytometry to detect apoptosis. we detected cell viability by CCK8, combined with the state of the nucleus after staining with Hoechst 33342 to determine apoptosis.
